# L-selectin mechanochemistry restricts neutrophil priming *in vivo*

Zhenghui Liu[1], Tadayuki Yago[1], Nan Zhang[2], Sumith R. Panicker[1], Ying Wang[2], Longbiao Yao[1], Padmaja Mehta-D'souza[1], Lijun Xia[1,2], Cheng Zhu[3,4,5] & Rodger P. McEver[1,2]

Circulating neutrophils must avoid premature activation to prevent tissue injury. The leukocyte adhesion receptor L-selectin forms bonds with P-selectin glycoprotein ligand-1 (PSGL-1) on other leukocytes and with peripheral node addressin (PNAd) on high endothelial venules. Mechanical forces can strengthen (catch) or weaken (slip) bonds between biological molecules. How these mechanochemical processes influence function *in vivo* is unexplored. Here we show that mice expressing an L-selectin mutant (N138G) have altered catch bonds and prolonged bond lifetimes at low forces. Basal lymphocyte homing and neutrophil recruitment to inflamed sites are normal. However, circulating neutrophils form unstable aggregates and are unexpectedly primed to respond robustly to inflammatory mediators. Priming requires signals transduced through L-selectin N138G after it engages PSGL-1 or PNAd. Priming enhances bacterial clearance but increases inflammatory injury and enlarges venous thrombi. Thus, L-selectin mechanochemistry limits premature activation of neutrophils. Our results highlight the importance of probing how mechanochemistry functions *in vivo*.

[1] Cardiovascular Biology Research Program, Oklahoma Medical Research Foundation, Oklahoma City, Oklahoma 73104, USA. [2] Department of Biochemistry and Molecular Biology, University of Oklahoma Health Sciences Center, Oklahoma City, Oklahoma 73104, USA. [3] Coulter Department of Biomedical Engineering, Georgia Institute of Technology, Atlanta, Georgia 30332, USA. [4] Woodruff School of Mechanical Engineering, Georgia Institute of Technology, Atlanta, Georgia 30332, USA. [5] Institute for Bioengineering and Bioscience, Georgia Institute of Technology, Atlanta, Georgia 30332, USA. Correspondence and requests for materials should be addressed to R.P.M. (email: rodger-mcever@omrf.org).

Mechanical forces regulate the lifetimes of reversible interactions (bonds) between biological molecules[1–3]. Force can destabilize interactions, shortening lifetimes (slip bonds). Counterintuitively, force can also stabilize interactions, prolonging lifetimes (catch bonds). Several studies have probed the structural basis for these mechanochemical processes. *In vitro*, force-regulated transitions between catch and slip bonds influence cell adhesion, morphology and signalling[1–3]. However, how mechanochemistry regulates physiological function *in vivo* is unexplored. Filling this major gap in knowledge is crucial, because extrapolations of *in vitro* functions to complex *in vivo* settings may be misleading or incomplete.

Under basal conditions, circulating neutrophils must remain quiescent. In response to infection or injury, they must adhere to vascular surfaces despite the forces exerted by blood flow[4,5]. Selectins initiate rolling adhesion of neutrophils in venules. Rolling neutrophils integrate signals that activate integrins, which slow rolling and mediate arrest and crawling. Neutrophil signalling must be regulated to permit adherent neutrophils to exit blood vessels before releasing reactive oxygen species (ROS), proteases, neutrophil extracellular traps (NETs), and other effectors that destroy pathogens[6,7]. Dysregulated signalling may prime neutrophils to release these effectors prematurely, injuring host tissues[8,9].

The three selectins mediate the first adhesive step during inflammation[4,5]. L-selectin is expressed on leukocytes, and P- and E-selectin are expressed on activated endothelial cells and/or platelets. Each selectin has an N-terminal C-type lectin domain, an epidermal growth factor (EGF)-like domain, a series of consensus repeats, a transmembrane domain and a cytoplasmic tail. Selectins mediate rolling through reversible interactions of their lectin domains with cell-surface glycosylated ligands. During inflammation, endothelial P- and E-selectin interact with P-selectin glycoprotein ligand-1 (PSGL-1) and other ligands on neutrophils. L-selectin also interacts with PSGL-1, enabling neutrophils to roll on already arrested neutrophils. In addition, L-selectin mediates lymphocyte homing to lymph nodes by interacting with peripheral node addressin (PNAd) on endothelial cells in high endothelial venules.

All selectins can transition between catch and slip bonds with their ligands as applied force changes[10–12]. *In vitro*, L-selectin catch bonds govern flow-enhanced adhesion, another counterintuitive phenomenon[13]. As flow increases from a threshold to an optimal value, force prolongs (catch) bond lifetimes to slow rolling. As flow increases above the optimum, force shortens (slip) bond lifetimes to accelerate rolling. *In vitro* studies support the hypothesis that catch bonds limit leukocyte adhesion to vessels during stasis and prevent aggregation of free-flowing leukocytes[14], which express both L-selectin and PSGL-1. Under both conditions, L-selectin bonds would have shorter lifetimes because little force is exerted on them. However, how L-selectin mechanochemistry acts *in vivo* has not been tested.

Crystal structures of each selectin[15–17] and molecular dynamics simulations[14,18] provide a rationale for using L-selectin as a molecular model to study the functions of catch and slip bonds *in vivo*. Force-regulated opening of a hinge between the lectin and EGF domains transmits allosteric change to the ligand-binding surface of the lectin domain, prolonging bond lifetimes (catch bonds)[14,19]. This information has guided studies of L-selectin mechanochemistry *in vitro*. The closed-hinge conformation of L-selectin is stabilized by a hydrogen bond between Tyr37 in the lectin domain and Asn138 in the EGF domain[16]. For L-selectin to open the hinge, the hydrogen bond must be disrupted. P-selectin has a Gly at residue 138 that does not interact with Tyr37, favouring hinge flexibility[15]. Therefore, higher forces are required

to induce catch bonds for L-selectin than P-selectin[10,11]. Substituting Gly for Asn138 in L-selectin (L-selectinN138G) reduces the force range for catch bonds and prolongs their lifetimes[14]. In addition, the more flexible hinge increases the on-rate for L-selectin interactions. These changes reduce the shear threshold for rolling but also increase aggregation between L-selectinN138G-expressing microspheres and PSGL-1-expressing neutrophils in a flow field[14].

To examine how altering mechanochemistry affects L-selectin function *in vivo*, we generated mice expressing L-selectinN138G. Dysregulating force-dependent lifetimes of L-selectin bonds modestly increases aggregation of circulating neutrophils. Unexpectedly, this dysregulation primes the neutrophils to react vigorously to subsequent signals, leading to enhanced bacterial clearance but also increased neutrophil-mediated tissue damage. Thus, L-selectin mechanochemistry regulates both cell adhesion and signalling to restrict inappropriate activation of circulating neutrophils.

## Results

**Enhanced neutrophil–neutrophil interactions in N138G mice.** We made knockin mice expressing L-selectinN138G on both alleles (Fig. 1a and Supplementary Fig. 1), herein termed N138G mice. Mice of both sexes were healthy and fertile like wild-type (WT) mice. Peripheral blood (PB) counts in N138G mice were normal (Supplementary Table 1). Histology of major organs was normal, as was cellularity of bone marrow (BM), thymus, spleen and lymph nodes (Supplementary Fig. 2a). Labelled splenocytes from N138G mice homed normally to lymph nodes (Supplementary Fig. 2b). BM neutrophils from N138G mice expressed slightly less L-selectin (Fig. 1b), slightly more integrin αMβ2 (Mac-1) (Fig. 1c), and normal levels of integrin αLβ2 (LFA-1) and several other glycoproteins (Supplementary Fig. 3).

To compare the force-regulated dissociation of L-selectin and L-selectinN138G bonds, we perfused BM leukocytes over low densities of 2-GSP-6, a glycosulfopeptide that mimics the N-terminal binding site for L-selectin on PSGL-1, or 6-sulfo-sLe$^x$, the glycan binding determinant for L-selectin on PNAd. The lifetimes of transient tethers, each containing most likely a single bond, were measured by high-speed videomicroscopy[10,11]. Lifetimes of L-selectin bonds with either ligand manifested a biphasic pattern characteristic of transitions from catch to slip bonds (Fig. 1d,e). As shear stress (applied force) increased, mean lifetimes increased until an optimal value was reached; further increases in force progressively shortened lifetimes. In contrast, lifetimes of L-selectinN138G bonds with each ligand were longer, particularly at low shear stresses, and exhibited slip bond behaviour at all shear stresses tested. Thus, the forces required to elicit catch bonds were too low to detect in this assay. This differs from studies of human L-selectinN138G in which the force range for catch bonds was detectable, albeit significantly reduced[14]. To determine whether altered bond lifetimes affected rolling, we perfused BM leukocytes over higher densities of 2-GSP-6 or 6-sulfo-sLe$^x$. Greater than 90% of rolling cells were neutrophils, as documented by nuclear morphology and Ly6G staining. Other BM cells expressed low levels of L-selectin and did not roll under these conditions. As shear stress increased, the rolling velocities of WT neutrophils first decreased, reached a minimum and then increased (Fig. 1f,g). In parallel, the number of rolling neutrophils first increased, reached a maximum and then decreased (Fig. 1h,i). However, there was no shear threshold for rolling of N138G neutrophils. Rolling velocities were much lower at low shear stresses and increased gradually as shear stress increased (Fig. 1f,g). Furthermore, the number of rolling neutrophils was maximal at the lowest shear stress and

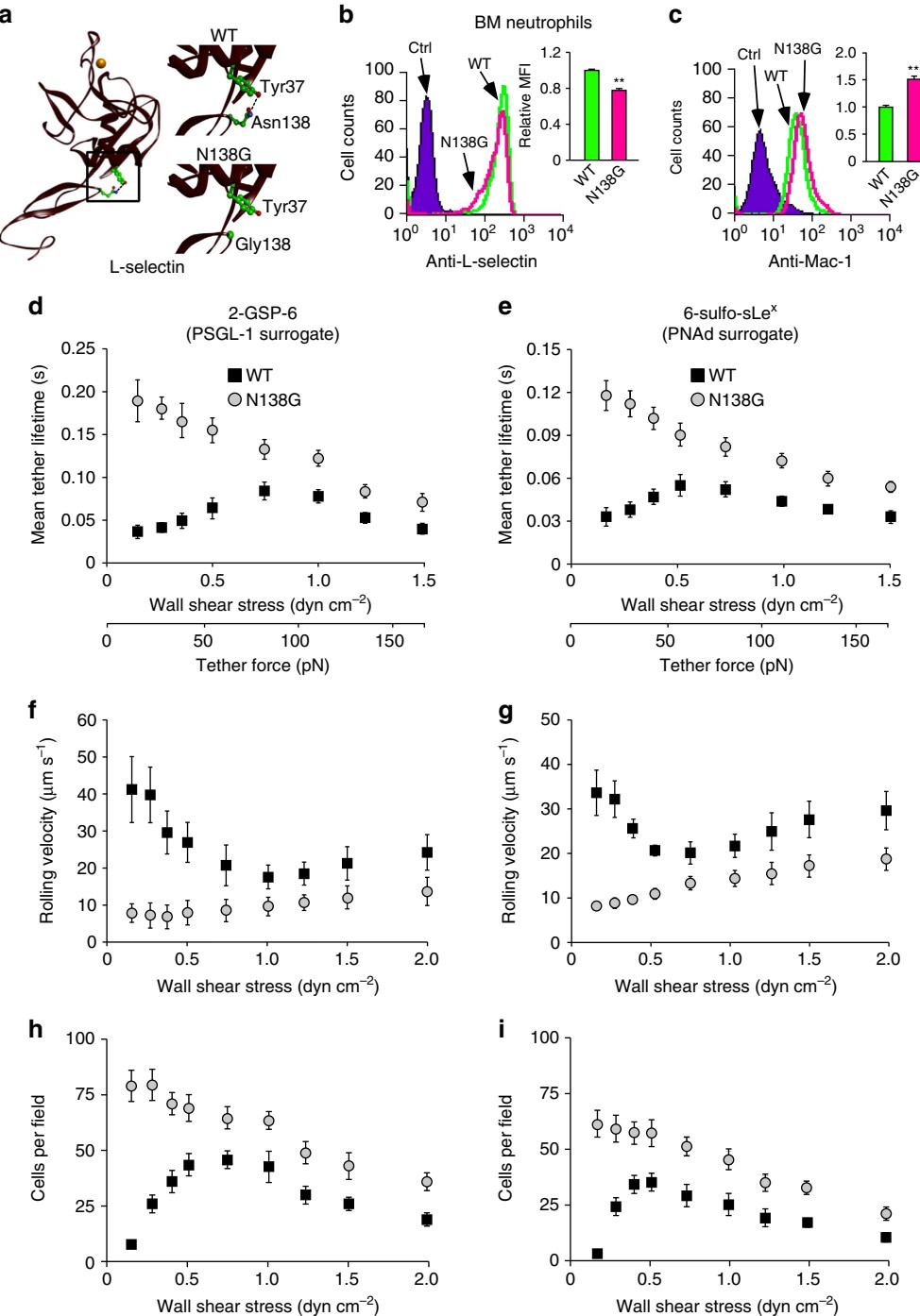

**Figure 1 | Altered L-selectin mechanochemistry in N138G mice reduces the shear threshold for neutrophil rolling.** (**a**) Left, crystal structure of the lectin and EGF domains of L-selectin (Protein Data Bank entry 3CFW). The gold sphere represents a $Ca^{2+}$ ion in the ligand-binding region of the lectin domain. The boxed area highlights the hinge region magnified in the insets (right). A hydrogen bond (dashed line) connects Tyr37 with Asn138. No hydrogen bond connects Tyr37 with Gly138 in the L-selectin mutant expressed in N138G mice. (**b,c**) Surface expression of L-selectin and Mac-1 on bone marrow (BM) neutrophils of WT and N138G mice. The gating strategy for BM neutrophils is shown in Supplementary Fig. 9. Representative histograms and relative mean fluorescence intensity (MFI) in bar graphs are shown. Data are mean ± s.e.m. of five mice per group. **P < 0.01 (two-tailed Student's t test). (**d–i**) Flow chamber assays of L-selectin-dependent interactions of BM neutrophils with 2-GSP-6 (surrogate for PSGL-1) (**d,f,h**) or with 6-sulfo-sLe$^x$ (surrogate for PNAd) (**e,g,i**). Transient tether lifetimes on lower ligand density (**d,e**), mean rolling velocities (**f,g**) and rolling cell numbers (**h,i**) on higher ligand density. The data represent the mean ± s.d. from five experiments.

gradually decreased with rising shear stress (Fig. 1h,i). Similar observations were made with splenocytes and PB neutrophils (Supplementary Fig. 4).

We asked whether longer L-selectin bond lifetimes at low forces enhance neutrophil–neutrophil interactions. Non-stirred BM neutrophils from N138G mice spontaneously aggregated in buffer after addition of $Ca^{2+}$ and $Mg^{2+}$, which are required for selectin and β2 integrin interactions (Fig. 2a). Anti-L-selectin mAb and anti-PSGL-1 mAb, but not isotype control mAb or anti-β2 integrin mAb, prevented aggregation, confirming its

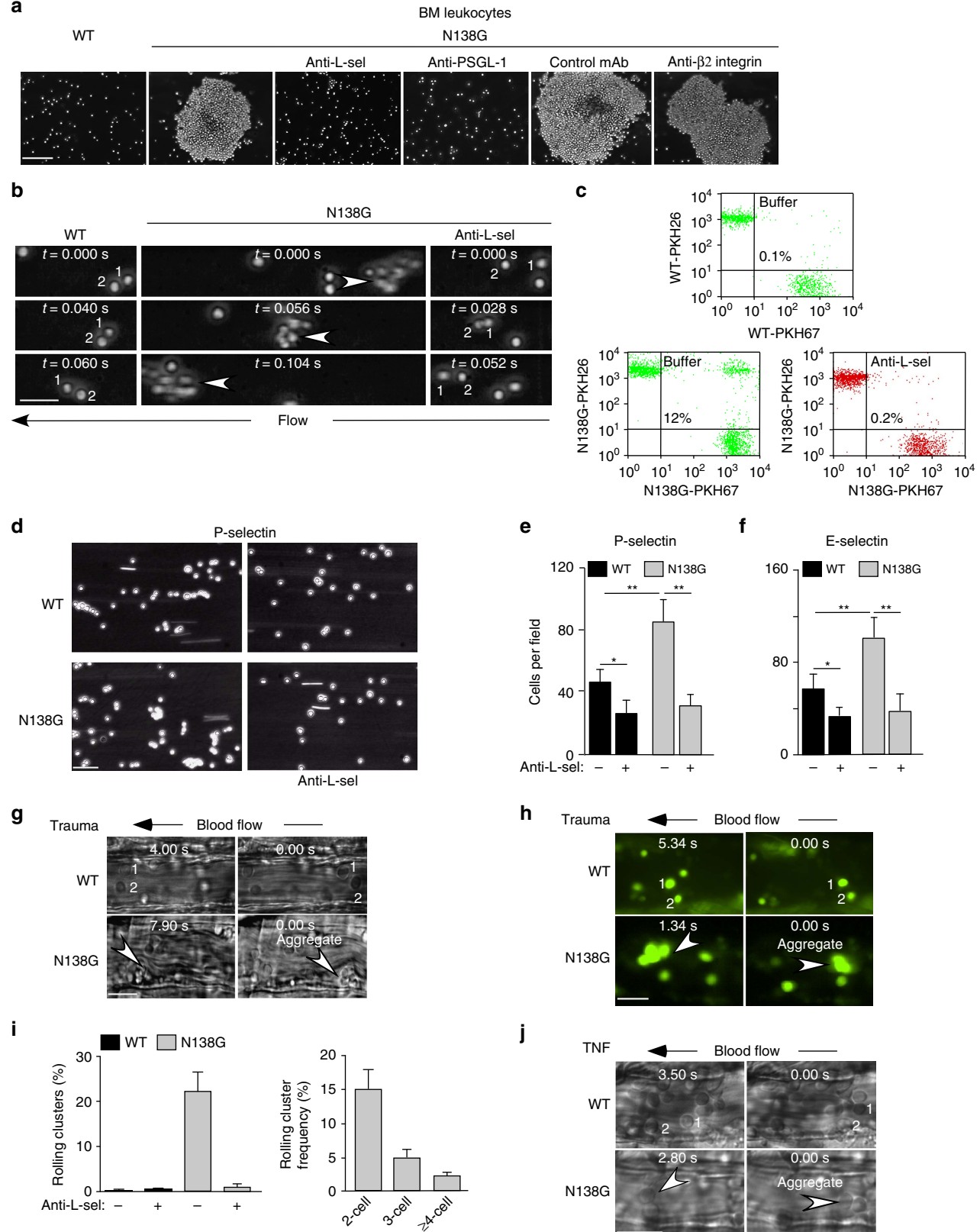

dependence on L-selectin-PSGL-1 interactions and indicating that, under these conditions, sufficient signalling did not occur to stabilize aggregates through integrin-dependent interactions. Similarly, flowing BM neutrophils from N138G but not WT mice formed aggregates that persisted until they left the field of

view (Fig. 2b). To quantify these interactions, we perfused 1:1 mixtures of BM leukocytes labelled with different fluorescent dyes and fixed the mixtures after they exited the flow chamber. Flow cytometry revealed that N138G but not WT neutrophils formed aggregates labelled with both dyes (Fig. 2c). Preincubation with

anti-L-selectin mAb prevented aggregation. *In vitro*, flowing neutrophils use L-selectin-PSGL-1 interactions to tether to and roll on neutrophils already rolling on immobilized P- or E-selectin[4]. These neutrophils transfer to the flow chamber surface to amplify the number of cells rolling on P- or E-selectin. BM neutrophils from N138G mice formed significantly more secondary interactions as they rolled on immobilized P- or E-selectin (Fig. 2d). Secondary interactions increased the total number of neutrophils rolling directly on P- or E-selectin (Fig. 2e,f). Anti-L-selectin mAb blocked all secondary interactions.

We used intravital microscopy to visualize neutrophil adhesion in postcapillary venules of the cremaster muscle after surgical trauma, which mobilizes P-selectin from Weibel-Palade bodies to the surfaces of endothelial cells. Most neutrophils rolled as single cells in WT mice, whereas a significant fraction of neutrophils rolled as small clusters in N138G mice (Fig. 2g). We crossed N138G mice with knockin LysM-GFP mice that express GFP only in myeloid cells. Fluorescence microscopy confirmed that virtually all cells in rolling aggregates were GFP-positive (Fig. 2h). Each cluster was counted as one rolling unit. Most N138G clusters were doublets and triplets, although aggregates as large as nine cells were observed (Fig. 2h,i). Rolling clusters were unstable, enlarging by adding other cells or dissociating into smaller clusters or single cells. Larger clusters broke up before permanently occluding venules. Injecting anti-L-selectin F(ab')$_2$ rapidly dissociated rolling aggregates into single cells (Fig. 2i); some single cells continued to roll whereas others detached into the flowing blood. Small, reversible L-selectin-dependent clusters of neutrophils also rolled in venules of the cremaster muscle after local injection of TNF, which upregulates expression of both P- and E-selectin (Fig. 2j). Thus, altering L-selectin mechanochemistry increases neutrophil aggregation *in vivo*. However, the aggregates are unstable and do not occlude blood vessels under the conditions studied.

**Circulating neutrophils in N138G mice are primed.** Unlike BM neutrophils, PB neutrophils from N138G mice expressed ∼60% less L-selectin (Fig. 3a), ∼3-fold more Mac-1 (Fig. 3b), and slightly less CD43, CD44, PSGL-1 and CD49d (Supplementary Fig. 5a). These altered surface markers suggest that N138G neutrophils are primed (minimally activated) after release from BM into the circulation[8,9]. Priming induces redistribution of Mac-1, but not LFA-1, from secretory vesicles to the plasma membrane[20]. PB neutrophils from N138G mice expressed normal levels of LFA-1 (Supplementary Fig. 5a). Neutrophil priming also induces partial, metalloprotease-mediated shedding of the ectodomains of L-selectin and some other glycoproteins[21]. An antibody to the cytoplasmic domain revealed normal L-selectin levels in permeabilized PB neutrophils from N138G mice, supporting retention of the transmembrane and cytoplasmic domains (Supplementary Fig. 5b). Scanning electron microscopy revealed fewer microvilli on PB but not BM neutrophils from N138G mice (Fig. 3c), consistent with partial actin rearrangements during priming[8,9]. Unlike aged circulating neutrophils[22], PB neutrophils from N138G mice did not express higher levels of CXCR4, CD11c or CD49d, and they did not express more ICAM-1, as do neutrophils stimulated with TNF or lipopolysaccharide[23] (Supplementary Fig. 5a). N138G mice expressed less L-selectin on lymphocytes and monocytes from PB and on lymphocytes from spleen and lymph nodes (Supplementary Fig. 5c), supporting priming of other leukocyte subsets in the circulation.

To determine whether priming of circulating N138G neutrophils affected integrin-dependent adhesion, we flowed BM or PB leukocytes over P- or E-selectin with or without coimmobilized ICAM-1. BM and PB neutrophils from WT and N138G mice rolled with similar velocities on P- or E-selectin and exhibited similar β2 integrin-dependent slow rolling on ICAM-1 (Supplementary Fig. 6a,b). A small number of PB neutrophils from N138G mice underwent β2 integrin-dependent arrest on ICAM-1 (Supplementary Fig. 6c,d). However, similar numbers of BM and PB neutrophils from both genotypes arrested on ICAM-1 when the chemokine CXCL1 was coimmobilized (Supplementary Fig. 6e,f). Likewise, a few neutrophils rolling on P-selectin in trauma-stimulated venules of N138G mice spontaneously arrested (Supplementary Fig. 6g). Nevertheless, the number of arrested neutrophils in TNF-stimulated venules, which express CXCL1 (ref. 24), did not increase in N138G mice (Supplementary Fig. 6h). These data suggest that priming of N138G neutrophils has minimal effect on β2 integrin-dependent adhesion under the conditions examined.

A hallmark of neutrophil priming with prototypical agonists such as TNF or GM-CSF is enhanced production of ROS in response to a second agonist[8,9]. Likewise, increasing concentrations of phorbol myristate acetate (PMA) induced significantly more ROS in PB neutrophils from N138G mice than from WT mice (Fig. 3d), whereas BM neutrophils from both genotypes produced equivalent ROS (Fig. 3e). However, priming of PB neutrophils in N138G mice was not accompanied by detectable systemic inflammation. A multiplex assay revealed normal plasma levels of 22 cytokines and chemokines, including IL-1β, TNF, GM-CSF and CXCL1, and an elevated level of IL-10, which typically exerts anti-inflammatory effects (Supplementary Table 2).

**Eliminating selectin ligands in N138G mice prevents priming.** Neutrophils rolling on P- or E-selectin transduce signals through PSGL-1 *in vitro* and *in vivo*[5,25]. Crosslinking L-selectin by antibodies or ligand mimetics initiates signalling *in vitro*[26–30], but whether L-selectin signals after engaging PSGL-1 or PNAd *in vivo* is unknown. We hypothesized that neutrophil priming in N138G mice resulted from longer L-selectin bond lifetimes that increased

**Figure 2 | Altered L-selectin mechanochemistry in N138G mice enhances neutrophil interactions *in vitro* and *in vivo*.** (a) Images of bone marrow (BM) leukocytes taken 30 min after resuspension in buffer containing Ca$^{2+}$ and Mg$^{2+}$ with or without blocking anti-L-selectin (L-sel) mAb, anti-PSGL-1 mAb, anti-β2 integrin mAb, or isotype control mAb. Bar, 100 μm. (b) Images of free-flowing BM leukocytes captured at 250 frames per second. Arrowheads mark movement of flowing cell aggregates. Bar, 20 μm. (c) Mixtures of BM leukocytes labelled with PKH26 and PKH67 dyes with or without anti-L-selectin mAb were fixed after exiting the flow chamber and analysed by flow cytometry. The percentage of aggregates labelled with both dyes is displayed in the top right quadrant. (d) Images of BM neutrophils with or without anti-L-selectin mAb rolling on P-selectin. Bar, 50 μm. (e,f) Number of BM neutrophils rolling on P- or E-selectin in the presence or absence of anti-L-selectin mAb. Data are mean ± s.d. from five experiments. *P<0.05; **P<0.01 (two-tailed Student's *t* test). (g) Images of individual leukocytes (numbered) or leukocyte clusters (arrowheads) rolling in venules of the cremaster muscle subjected to trauma from a WT or N138G mouse. Bar, 25 μm. (h) Fluorescent images of GFP-positive individual neutrophils (numbered) or neutrophil clusters (arrowheads) rolling in a venule of the cremaster muscle subjected to trauma from a WT/LysM-GFP or N138G/LysM-GFP mouse. Bar, 20 μm. (i) Quantification of rolling clusters and cluster size in venules of the cremaster muscle subjected to trauma from WT or N138G mice. Data are mean ± s.e.m. for 26–30 venules from eight mice for each group. (j) Images of individual leukocytes (numbered) or leukocyte clusters (arrowheads) rolling in venules of the cremaster muscle stimulated with TNF from a WT or N138G mouse. Bar, 25 μm. The data in **a–d,g,h** and **j** are representative of three experiments.

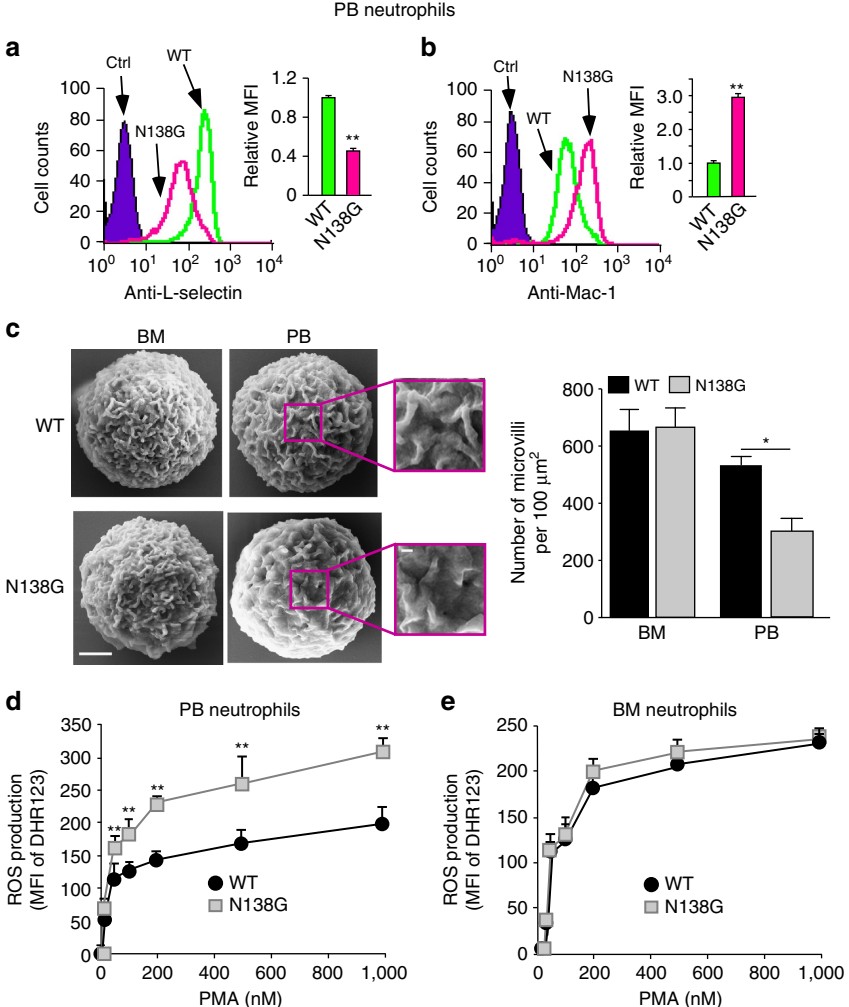

**Figure 3 | Circulating neutrophils in N138G mice are primed.** (**a**,**b**) Surface expression of L-selectin and Mac-1 on peripheral blood (PB) neutrophils of WT and N138G mice. Representative histograms and relative mean fluorescence intensity (MFI) in bar graphs are shown. The gating strategy for PB neutrophils is shown in Supplementary Fig. 9. Data are mean ± s.e.m. of five mice per group. (**c**) Scanning electron microscopy of neutrophils isolated from PB or BM of WT and N138G mice (Bar, 1 μm). The boxed areas are magnified in the insets. Quantification of cell-surface microvilli in bar graph. Data are mean ± s.e.m. of 15–20 neutrophils from five mice per genotype. (**d**,**e**) Reactive-oxygen species (ROS) generated by PB or BM neutrophils in response to phorbol myristate acetate (PMA). Data are mean ± s.e.m. from six mice per group in three experiments. *$P < 0.05$; **$P < 0.01$ (two-tailed Student's $t$ test).

signalling through L-selectin and/or PSGL-1. To test this hypothesis, we asked whether deleting selectin ligands in N138G mice would prevent shedding of L-selectin and upregulation of Mac-1 on PB neutrophils. To bind selectins, glycans on PSGL-1, PNAd and other ligands must be α1-3-fucosylated by fucosyltransferases IV and VII (FTIV and FTVII)[31]. Crossing N138G mice with double knockout mice lacking both FTIV and FTVII (FT DKO) to disable selectin binding restored normal levels of L-selectin and Mac-1 on PB neutrophils (Fig. 4a,b), and normalized ROS production in response to PMA (Fig. 4c). We next crossed N138G mice with PSGL-1-deficient mice[32] or with lymphotoxin-α-deficient mice, which lack almost all lymph nodes and therefore express little or no PNAd[33]. Progeny of both crosses failed to restore normal L-selectin and Mac-1 levels on PB neutrophils (Fig. 4d–g). Transplantation of N138G/LysM-GFP BM cells into lethally irradiated FT DKO mice also failed to rescue priming (Fig. 4h). These data indicate that L-selectin engagement with either PSGL-1 or PNAd is sufficient to prime circulating neutrophils in N138G mice. The retention of priming in N138G mice lacking PSGL-1 suggests that neutrophils rolling on PNAd transduce signals directly through L-selectin. To further test this hypothesis, we exploited mixed radiation chimeras that expressed both donor (GFP-positive) and recipient (GFP-negative) haematopoietic cells. When N138G/LysM-GFP neutrophils circulated with WT or FT-DKO neutrophils, only the N138G/LysM-GFP neutrophils were primed (Fig. 4i, left two panels). When N138G/LysM-GFP neutrophils circulated with N138G/FT DKO neutrophils, both were primed (Fig. 4i, right panel). These data demonstrate that priming of N138G neutrophils requires signals transmitted through L-selectin rather than through PSGL-1.

**Neutrophil priming in N138G mice enhances bacterial killing.** To determine the pathophysiological consequences of neutrophil priming in N138G mice, we first asked whether it enhances responses to bacterial infection. N138G neutrophils in whole blood generated more ROS when exposed to *E. coli* (Fig. 5a), and they phagocytosed more fluorescently labelled *E. coli* (Fig. 5b). Equivalent numbers of neutrophils migrated into the peritoneum of WT and N138G mice 4 h after intraperitoneal injection of *E. coli* (Fig. 5c). At this time point, however, N138G mice, unlike WT mice, had cleared virtually all the bacteria from the peritoneum (Fig. 5d), and unlike WT mice, they did not become hypothermic, a hallmark of severe sepsis (Fig. 5e).

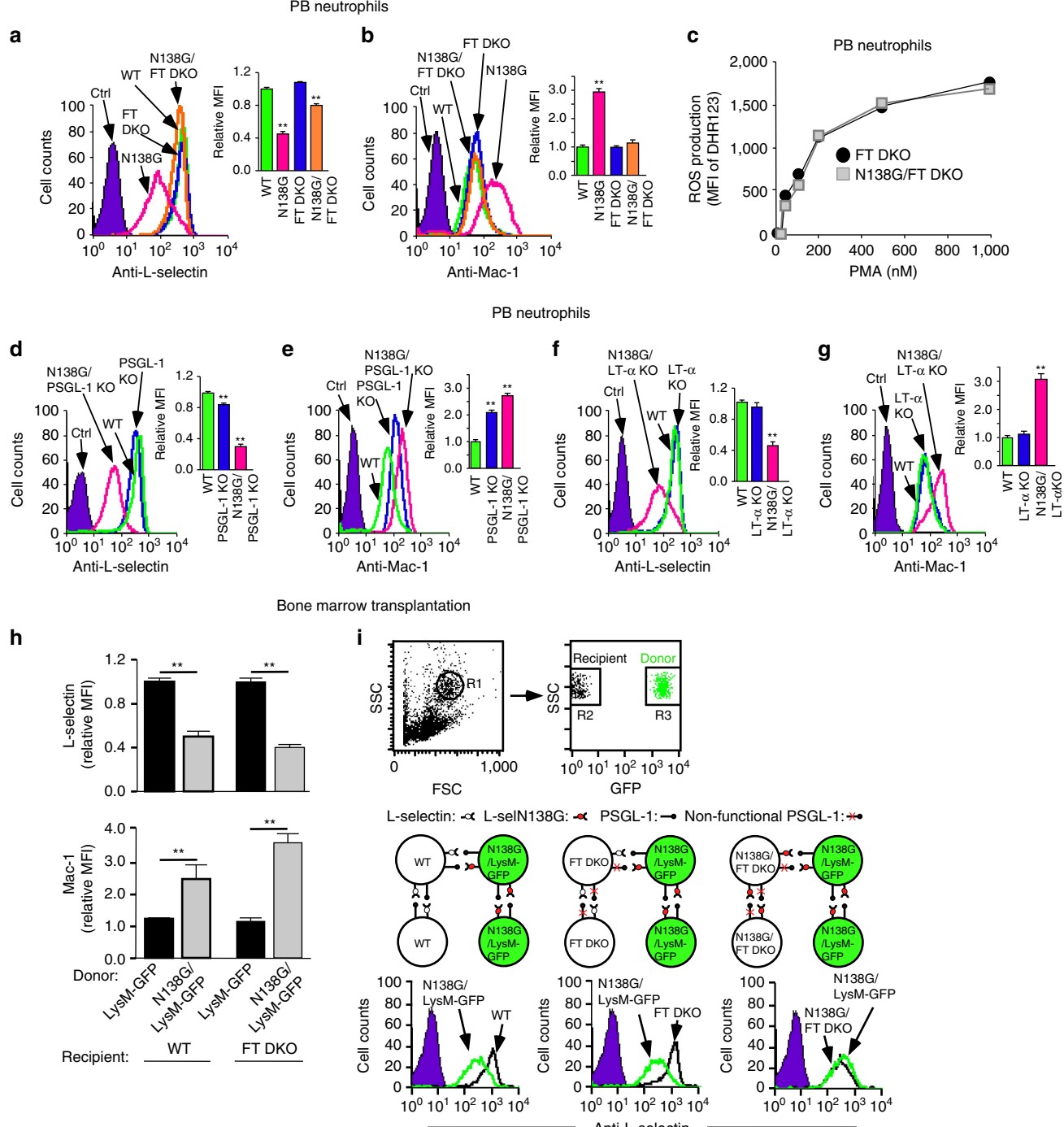

**Figure 4 | Eliminating selectin ligands in N138G mice prevents priming of circulating neutrophils.** (**a,b,d–g**) Surface expression of L-selectin and Mac-1 on PB neutrophils of the indicated genotype. Representative histograms and relative mean fluorescence intensity (MFI) in bar graphs are shown. The gating strategy for PB neutrophils is shown in Supplementary Fig. 9. (**c**) ROS generated by PB neutrophils from FT DKO or N138G/FT DKO mice in response to PMA. (**h**) Surface expression of L-selectin and Mac-1 on PB neutrophils from irradiated WT or FT DKO mice reconstituted with donor BM from LysM-GFP or N138G/LysM-GFP mice. (**i**) Surface expression of L-selectin on PB neutrophils from irradiated WT, FT DKO or N138G/FT DKO mice (black) partially reconstituted with donor BM from N138G/LysM-GFP mice (green). Top, PB neutrophil populations gated by forward and side scatter (R1) were analysed for fluorescence to distinguish GFP-negative recipient cells (R2) from GFP-positive donor cells (R3). Bottom, binding schematics and representative histograms. The data in **a–h** are mean ± s.e.m. of five mice per group. The data in **i** are representative of three mice per group. **P < 0.01 (two-tailed Student's *t* test).

**Neutrophil priming in N138G mice slows healing after injury.** We then explored whether neutrophil priming in N138G mice has adverse effects. WT and N138G mice recruited similar numbers of neutrophils into the peritoneum after challenge with thioglycollate (Fig. 6a) or into an air pouch containing CXCL1 or CXCL2 (Fig. 6b). In both models, emigrated neutrophils in N138G mice retained the ability to generate more ROS in response to PMA (Fig. 6c,d). WT and N138G mice also mobilized

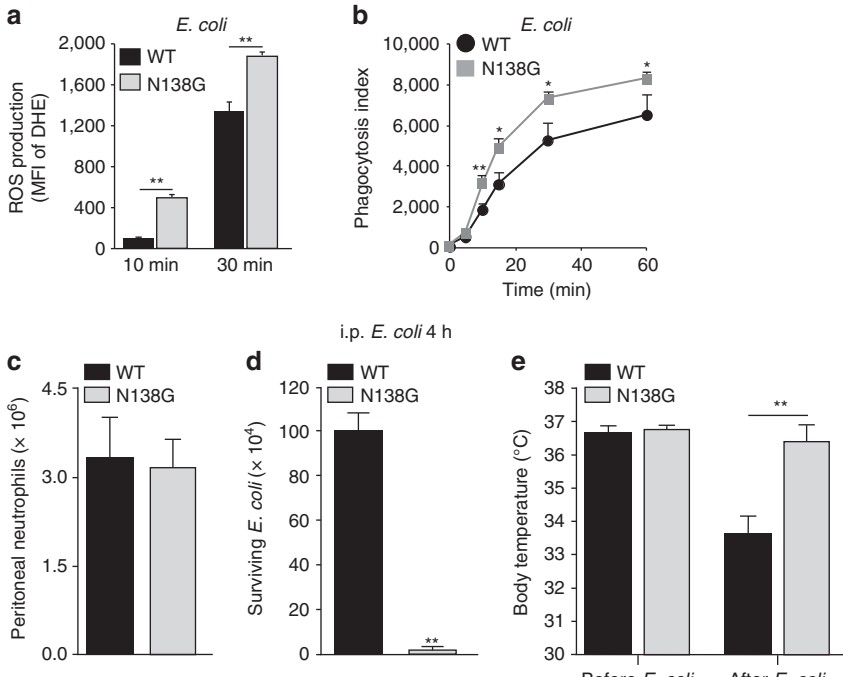

**Figure 5 | Primed neutrophils from N138G mice have enhanced bacterial killing *in vitro* and *in vivo*.** (**a**) ROS generated by neutrophils in heparinized blood after incubation with *E. coli* for 10 or 30 min. (**b**) Phagocytosis of fluorescent *E. coli* by neutrophils in heparinized blood. (**c,d**) Number of neutrophils and bacteria recovered by peritoneal lavage 4 h after intraperitoneal injection of *E. coli*. (**e**) Body temperatures of mice before and 4 h after intraperitoneal injection of *E. coli*. Data in **a** and **b** are mean ± s.e.m. of four experiments with three mice per group. Data in **c–e** are mean ± s.e.m. of ten mice per group. *$P < 0.05$; **$P < 0.01$ (two-tailed Student's *t* test).

similar numbers of neutrophils into ears painted with croton oil (Fig. 6e,f). Nevertheless, ear swelling was much more prominent and prolonged in N138G mice (Fig. 6e,g). Thus, emigration of primed neutrophils delays healing after sterile injury.

**Neutrophil priming in N138G mice augments venous thrombosis.** Deep vein thrombosis is an inflammatory disorder in humans and other mammals, including mice. In mouse models of reduced flow in the inferior vena cava (IVC), thrombus formation is propagated by adhesive and signalling events among endothelial cells, neutrophils, monocytes and platelets[34,35]. Endothelial P-selectin recruits neutrophils to sites of nascent thrombi, but a function for L-selectin has not been examined. We hypothesized that N138G mice would form larger thrombi due to L-selectin-dependent aggregation and/or priming of myeloid cells. Priming could increase procoagulant activity through monocyte expression of tissue factor or neutrophil release of NETs[34]. Indeed, the plasma of N138G mice had increased tissue factor-dependent coagulant activity, consistent with circulating tissue factor-expressing microparticles derived from monocytes[36] (Supplementary Fig. 7). To further test this hypothesis, we ligated the IVC of WT or N138G mice to reduce blood flow by ~90%. After 48 h, the frequency of thrombus formation was similar in both genotypes, and was not affected by pre-injection of anti-L-selectin F(ab')$_2$ (Fig. 7a). However, N138G mice formed significantly larger thrombi (Fig. 7b,c). Anti-L-selectin F(ab')$_2$ did not affect the size of thrombi in WT mice but reduced the size of thrombi in N138G mice to WT levels (Fig. 7c). Immunostaining of thrombi indicated that most DAPI-positive nucleated cells were Ly6G-positive neutrophils in both genotypes (Fig. 7d). Stitching of cryosections revealed significantly fewer Ly6G-positive neutrophils relative to thrombus area in N138G mice (Fig. 7e,f), and western blots of thrombus lysates revealed less Ly6G protein relative to fibrin in N138G mice (Fig. 7g).

Anti-L-selectin F(ab')$_2$ did not affect Ly6G ratios in WT mice but restored those in N138G mice to WT levels (Fig. 7g). These data demonstrate that L-selectin-dependent priming rather than aggregation of myeloid cells augments venous thrombi in N138G mice.

## Discussion

We have shown that altering the mechanochemistry of an adhesion receptor profoundly affects its functions *in vivo*. The N138G substitution in L-selectin reduces the force range for transitions from catch to slip bonds and prolongs bond lifetimes at low forces. The phenotype of mice expressing this L-selectin mutant reveals unexpected insights into leukocyte adhesion and signalling that *in vitro* experiments did not anticipate. Thus, our data highlight the importance of probing mechanochemistry *in vivo*.

*In vitro*, the N138G mutation eliminated the shear threshold for rolling on L-selectin ligands. Rolling velocities were only slightly slower at physiological shear stresses above the threshold. This likely explains the normal L-selectin-dependent lymphocyte homing observed in N138G mice. *In vitro*, N138G neutrophils formed more L-selectin-dependent secondary interactions with neutrophils rolling on P- or E-selectin. *In vivo*, secondary tethers may be more important when neutrophils adhere in larger venules[37,38]. If more tethers formed in N138G mice, they did not increase neutrophil recruitment in our models of inflammation.

Based on our *in vitro* data, we predicted that N138G mice would have severe and perhaps lethal aggregation of circulating leukocytes. However, aggregation *in vivo* was surprisingly mild before or after inflammatory challenge, even in our restricted-flow model of deep vein thrombosis. Circulating L-selectin-dependent neutrophil aggregates were small and rapidly reversible. Probably too few β2 integrins were activated to stabilize the aggregates, despite priming (see below). *In vitro*, we observed

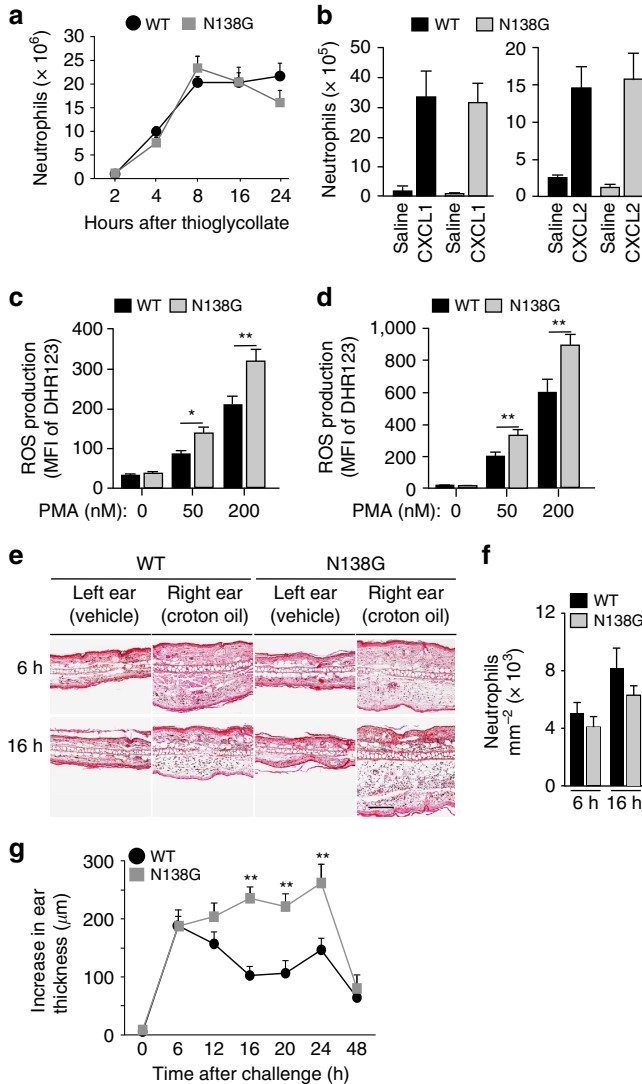

**Figure 6 | Neutrophil priming in N138G mice delays healing after sterile injury.** (**a**) Number of neutrophils recovered by lavage from the peritoneum at the indicated time after intraperitoneal injection of thioglycollate. (**b**) Number of neutrophils recovered by lavage from air pouches 4 h after injection with saline, CXCL1 or CXCL2. (**c**) ROS generated in response to PMA by peritoneal neutrophils recovered 4 h after injection of thioglycollate. (**d**) ROS generated in response to PMA by air-pouch neutrophils recovered 4 h after injection of CXCL1. (**e**) Representative haematoxylin/eosin-stained ear sections 6 h or 16 h after topical application of vehicle or croton oil. Bar, 200 μm. (**f**) Number of neutrophils, quantified by myeloperoxidase staining, in ear sections 6 h or 16 h after topical application of croton oil. (**g**) Increase in ear thickness at the indicated time after application of croton oil. Data are mean ± s.e.m. of four experiments with three mice per group. *$P < 0.05$; **$P < 0.01$ (two-tailed Student's $t$ test).

massive aggregation of BM leukocytes suspended in buffer. *In vivo*, the large excess of erythrocytes in blood may reduce leukocyte contacts. Moreover, naïve lymphocytes, which comprise ∼70% of circulating leukocytes in mice, express under-glycosylated PSGL-1 that does not bind to selectins[39]. The shedding of ∼60% of surface L-selectin might further limit agglutination of circulating cells, although sufficient L-selectin remained to eliminate the shear threshold for rolling *in vitro* and to support lymphocyte homing *in vivo*. It remains possible that some pathological states elicit harmful leukocyte aggregation in

N138G mice. Analogous to L-selectin, glycoprotein Ibα uses catch bonds to mediate flow-enhanced rolling of platelets on immobilized von Willebrand factor (VWF) but avoid agglutination of circulating platelets with plasma VWF[40]. Some VWF mutations abrogate catch bonds by prolonging glycoprotein Ibα-VWF bond lifetimes at low forces, and increase agglutination of flowing platelets with VWF-coated beads[40]. These mutations occur in type 2B von Willebrand disease, which is associated with thrombocytopenia, loss of plasma VWF multimers and bleeding due to VWF-mediated platelet agglutination[41]. In this case, the clinical findings are consistent with the *in vitro* observations. However, the surprisingly mild leukocyte aggregation in N138G mice illustrates how *in vitro* data can incorrectly predict *in vivo* outcomes.

Crosslinking L-selectin with antibodies or multivalent ligands propagates a signalling cascade *in vitro*[26–30], but the *in vivo* relevance of these studies was untested. Unexpectedly, the most dramatic phenotype in N138G mice was inappropriate signalling in leukocytes after they entered the circulation. Although we focused on neutrophils, all circulating leukocytes were primed. Neutrophils expressed lower surface L-selectin and higher surface Mac-1, indicators of increased basal stimulation, and they responded more robustly to a second stimulus, a key feature of priming[8,9]. Surprisingly, priming required signals transduced through L-selectinN138G but not PSGL-1, even though neutrophil aggregates could potentially signal bidirectionally. The persistence of priming in neutrophils that express L-selectinN138G but not PSGL-1 excludes a proposed L-selectin signalling mechanism that requires PSGL-1 to associate with L-selectin[42]. Our results support distinct requirements for signalling through L-selectin and PSGL-1. Both glycoproteins are concentrated in the tips of leukocyte microvilli[4]. However, PSGL-1 partitions in lipid rafts and requires intact rafts to signal[43], whereas L-selectin is distributed in non-raft membrane domains[44]. Signalling through PSGL-1 has been extensively documented *in vitro* and *in vivo*[5,25]. Signalling begins when PSGL-1 on neutrophils reversibly interacts with P- or E-selectin on activated platelets and/or endothelial cells. At all force levels, the lifetimes of P- and E-selectin bonds are 4–5-fold longer than those of L-selectin bonds[10–13]. The lifetimes of L-selectinN138G bonds, although longer at low forces than those of L-selectin bonds, are still significantly shorter than those of P- and E-selectin bonds[14]. In other molecular interactions, force prolongs the lifetimes of catch bonds that individually[45] or cumulatively[46] reach a signalling threshold. The signalling threshold for PSGL-1 may require the longer lifetimes of bonds with P- or E-selectin, which neither L-selectin nor L-selectinN138G can achieve. On the other hand, our data show that L-selectin signalling is sensitive to even small increases in lifetimes of bonds with PSGL-1 and PNAd, as seen in N138G mice. Thus, L-selectin mechanochemistry is precisely tuned to limit priming of circulating leukocytes.

Priming of neutrophils in N138G mice had little or no effect on rolling on P- or E-selectin or on integrin-dependent slow rolling or arrest on ICAM-1 *in vitro* or *in vivo*. Accordingly, N138G mice exhibited normal neutrophil recruitment after inflammatory challenge. Migrating neutrophils are thought to delay release of ROS, proteases, NETs and other effectors until they reach extravascular pathogens[6,7]. *In vitro*, primed N138G neutrophils generated more ROS and phagocytosed more *E. coli*. Remarkably, N138G mice rapidly cleared injected *E. coli* from the peritoneum, which prevented hypothermia, a hallmark of sepsis. These beneficial effects of neutrophil priming were accompanied by adverse side effects. Healing after sterile injury to the ear was delayed. Furthermore, N138G mice developed larger thrombi in the IVC after flow reduction. The shorter plasma clotting

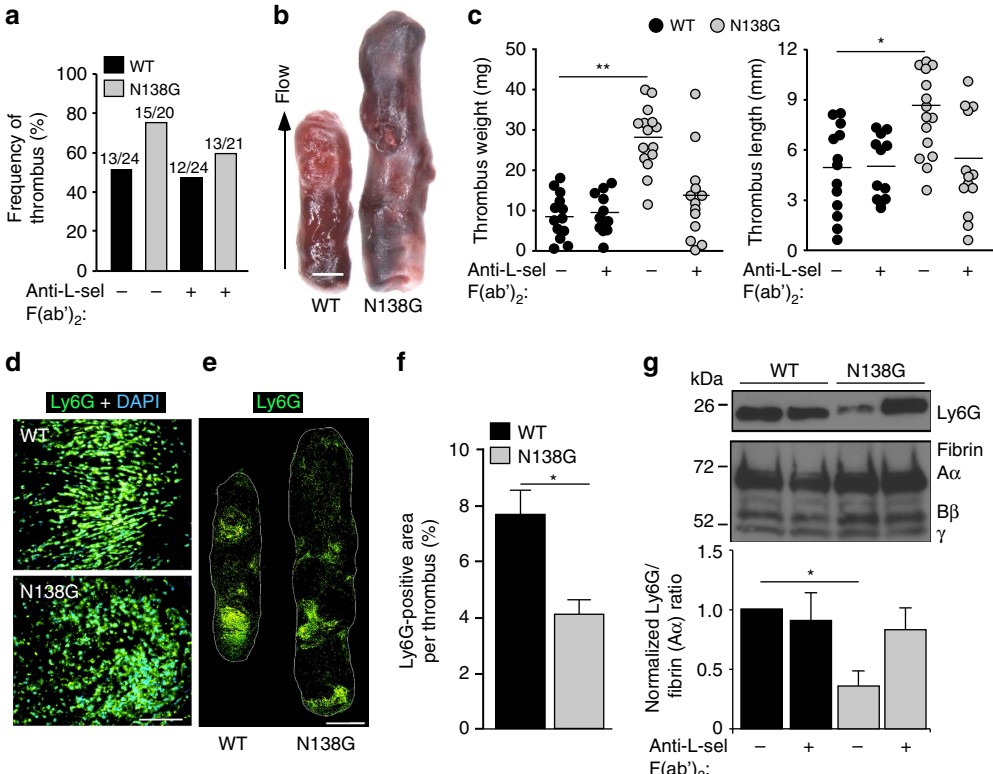

**Figure 7 | N138G mice form larger deep vein thrombi.** (**a**) Frequency of thrombi formed in inferior vena cava 48 h after flow restriction in WT and N138G mice treated with or without anti-L-selectin F(ab′)₂. For each experimental group, the number of mice forming thrombi relative to the total number of mice is shown. (**b**) Representative thrombi. Arrow indicates direction of blood flow. Bar, 1 mm. (**c**) Thrombus length and weight in WT and N138G mice treated with or without anti-L-selectin F(ab′)₂. Each symbol represents an individual thrombus. Horizontal bars represent median values. (**d**) Representative cryosections of leukocyte-rich areas of thrombi stained with anti-Ly6G mAb to identify neutrophils and with DAPI to identify cell nuclei. Bar, 100 μm. (**e**) Stitched cryosections of entire thrombus stained with anti-Ly6G mAb. Bar, 1 mm. (**f**) Percentage of Ly6G-positive area per thrombus area in stitched sections. Data are mean ± s.e.m. from seven mice per group. (**g**) Upper panel, representative western blot of thrombus lysates probed with antibodies to Ly6G and fibrin. The Aα, Bβ and γ chains of fibrin are marked. The full-length blot is shown in Supplementary Fig. 8. Lower panel, normalized densitometric ratio of Ly6G to fibrin Aα chain. Data are mean ± s.d. from eight mice per group. $*P < 0.05$; $**P < 0.01$ (two-tailed Student's $t$ test).

times in unchallenged N138G mice suggest that monocytes inappropriately expressed tissue factor, a cofactor for thrombin-dependent fibrin formation in venous thrombi[34]. Primed neutrophils in thrombi of N138G mice likely also generated more thrombogenic NETs[34]. Anti-L-selectin F(ab′)₂ reduced thrombus size in N138G mice, probably by preventing priming of myeloid cells that entered the circulation after venous ligation. Our results illustrate how L-selectin mechanochemistry helps to balance signals in myeloid cells to allow pathogen clearance and wound healing without collateral injury. Further studies will reveal whether other leukocyte subsets use L-selectin mechanochemistry to regulate innate or adaptive immunity.

Our methods provide a benchmark for exploring the functions of mechanochemistry *in vivo*. Similar approaches can be extended to diverse adhesion receptors, cytoskeletal proteins and other molecules influenced by physical forces.

## Methods

**Reagents.** Rat anti-mouse L-selectin mAb Mel-14 (rat IgG2a) was purified from hybridoma supernatants (American Type Culture Collection). Mel-14 F(ab′)₂ fragments were prepared using immobilized Pepsin according to the manufacturer's protocol (Pierce F(ab′)₂ Preparation Kit, #44988, Thermo Scientific). Goat anti-rat Fc IgG linked to agarose (AminoLink Plus Coupling Resin Column, #1856083, Thermo Scientific) was used to remove Fc fragments and undigested IgG. Rat anti-mouse E-selectin mAb 9A9, anti-mouse P-selectin mAb RB40.34 and anti-mouse PSGL-1 mAb 4RA10 have been described[47–49]. The following rat mAbs against mouse proteins were purchased from BD Biosciences: CD16/32 (2.4G2), PE- or FITC-Ly6G (1A8), FITC-LFA-1 (M17/4), FITC-Mac-1 (M1/70), β2 integrin (GAME46), FITC-Mel-14, FITC-CD43 (Ly-48), FITC-CD44 (IM7), FITC-PSGL-1

(2PH1), PE-CD115 (AFS98), PE-ICAM-1 (YN1/1.7.4), PE-CXCR4 (2B11), PE-CD49d (R1-2). PE-CD11c (N418) was purchased from eBioscience. Rabbit anti-human myeloperoxidase IgG (cross-reacts with mouse myeloperoxidase) was from Thermo Scientific. Alexa Fluor 488-conjugated donkey anti-rat IgG was from Invitrogen. Goat horseradish peroxidase-conjugated anti-rabbit IgG was from Cell Signaling Technology. Goat anti-human IgM Fc antibody was from Caltag. Goat anti-human IgG Fc antibody was from Chemicon.

**Mice.** All experiments with mice were approved by the Institutional Animal Care and Use Committee of the Oklahoma Medical Research Foundation. All mice were aged 8–12 weeks, except in deep vein thrombosis experiments, where mice were aged 12–16 weeks. Mice in experimental groups were matched for age and gender.

Knockout mice lacking PSGL-1 ($Selpl^{-/-}$, termed PSGL-1 KO) and homozygous knockin mice expressing GFP under control of the *LysM* promoter (LysM-GFP) have been described[32,50]. Knockout mice lacking α1-3-fucosyltransferases IV and VII ($Fut4^{-/-}/Fut7^{-/-}$, termed FT DKO)[31] were provided by Dr Jonathon Homeister (University of North Carolina). Knockout mice lacking lymphotoxin-α ($Lta^{-/-}$)[33] were provided by Dr Nancy Ruddle (Yale University). All mice were backcrossed at least ten generations in the C57BL/6J background. C57BL/6J mice and EIIa-Cre [Tg(EIIa-Cre)C5379Lmgd/J] mice were obtained from the Jackson Laboratory.

Knockin mice expressing L-selectinN138G on both alleles (N138G mice) were generated by previously characterized methods[32]. Briefly, an Entrez Gene search of GenBank revealed an 18-kb intact *Sell* genomic sequence. Exon 4 encodes Asn138 that is conserved in human L-selectin. The sequence around exon 4 was amplified by high-fidelity PCR using genomic DNA from CJ7 mouse embryonic stem cells. This sequence was used to prepare a knockin targeting vector. Site-directed mutagenesis was used to substitute three nucleotides in the codon for Asn138 so that it encoded Gly138. To avoid potential fatal complications from expression of the mutant gene during development, we introduced a loxP-flanked *neostop* cassette fragment (a gift from Dr Ioannis Dragatsis, University of Tennessee Health Science Center) into intron 2. In the presence of Cre recombinase, the floxed

cassette is deleted, allowing expression of the gene. Finally, a thymidine kinase (*tk*) gene was inserted adjacent to the 3′-flanking homologous sequence of the targeting vector to enrich for targeted clones by negative selection with ganciclovir. The fidelity of the targeting construct was verified by DNA sequencing. The linearized targeting vector was electroporated into CJ7 mouse ES cells. After G418 (300 μg ml$^{-1}$) and ganciclovir (2 μM) selection, surviving clones with correct homologous recombination were identified by PCR and confirmed by Southern blotting and DNA sequencing. Heterozygous ES cells from one clone confirmed to have a normal karyotype were microinjected into C57BL/6J blastocysts to generate chimeric mice. Germline transmission was achieved from two chimeric males. The 138G mutation was identified by PCR amplification of a 323-bp fragment from mouse tail genomic DNA, with the following primers: F—5′-GTGGAGAATGT-GTGGAAACTATCGGA-3′; R—5′-GGAGTTGAGTCTTGTTTCCGCTACTT-3′. Heterozygous mice were mated with EIIa-Cre mice. The EIIa-*cre* transgene, under the control of the adenovirus EIIa promoter, targets expression of Cre recombinase at the one-cell zygote stage and thus deletes the *neostop* cassette early during development. Heterozygous progeny that displayed complete *neostop* excision were bred to obtain homozygous F2 mice and WT littermate controls. F2 offspring without the EIIa-*cre* transgene were used for experiments and for further crossing with other strains. Littermate controls were used in all experiments with mice of mixed genetic background. The F2 offspring were also backcrossed at least ten generations in the C57BL/6J background and used in some experiments to confirm results. PCR and DNA sequencing were used to confirm the transmission of the N138G mutation.

**Flow cytometry and blood counts.** For flow cytometry, $1–2 \times 10^6$ freshly isolated bone marrow cells in 200 μl buffer or 100 μl whole blood were stained with conjugated antibodies on ice for 30 min. Fc receptors were blocked by incubation with anti-mouse CD16/CD32 antibodies for 5 min on ice. After antibody incubation, erythrocytes were lysed by adding 2 ml lysis buffer (150 mM NH$_4$Cl, 10 mM NaHCO$_3$, 1 mM EDTA) for 3 min at room temperature. After a final wash, samples were analysed on a FACSCalibur cytometer (Becton Dickinson). Leukocyte subpopulations were distinguished by light scatter and/or by specific antibodies. Cold Ca$^{2+}$/Mg$^{2+}$-free Hanks' balanced salt solution containing 5 mM EDTA and 0.1% human serum albumin (HBSS$^-$/EDTA/HSA) was used unless otherwise specified. Peripheral blood counts were measured using a veterinary haematology analyser (Hemavet 950, Drew Scientific Inc.).

**Scanning electron microscopy.** Mouse blood collected in 10 mM EDTA by heart puncture was immediately fixed in 4% formaldehyde for 10 min. After lysing erythrocytes, leukocytes were centrifuged and stained with anti-mouse Ly6G-PE (5 μg ml$^{-1}$) in 200 μl HBSS$^-$/EDTA/HSA. Neutrophils were sorted on a Beckman Coulter MoFlo XDP instrument based on light scatter and staining for Ly6G-PE. The sorted cells (>97% pure) were resuspended in 4% glutaraldehyde (Electron Microscopy Sciences) and settled on 0.001% polylysine-coated coverslips overnight. The samples were washed with sodium cacodylate (0.1 M, pH 7.4) and incubated with 1% osmium tetroxide (Electron Microscopy Sciences) for 1 h. After washing with distilled water, samples were dehydrated through a graded ethanol series (25–100%). The samples were incubated with hexamethyldisilazane (Sigma-Aldrich) followed by air-drying. Finally, samples were sputter coated with 4-nm iridium using an Emitech K575D instrument, and cells were viewed with a Zeiss Neon-40EsB scanning electron microscope under 5-kV accelerating voltage at the Samuel Roberts Noble Microscopy Laboratory at the University of Oklahoma in Norman. A microvillus was defined as any membrane fold that projected from the cell surface with a ridge and trough. To quantify the number of microvilli per unit area, a 1-μm$^2$ box was drawn at the centre of each cell. Microvilli were counted and expressed as average number per 100 μm$^2$.

**Lymphocyte homing.** Short-term homing of lymphocytes was conducted as described[51]. Briefly, splenocytes were suspended in HBSS$^-$/EDTA/HSA. After lysis of erythrocytes in 150 mM NH$_4$Cl, 10 mM NaHCO$_3$, 1 mM EDTA, splenocytes were washed, counted and labelled with either CMFDA or CMTMR (Life Technologies) following the vendor's recommendations. Cells were washed twice, counted and resuspended in saline at $10^8$ per ml. $10^7$ CMFDA-labelled splenocytes from WT or N138G mice mixed with $10^7$ CMTMR-labelled WT splenocytes were injected intravenously into WT mice. An aliquot of mixtures was used to assess the input ratio calculated as CMFDA-positive cells/CMTMR-positive cells. After 2 h, the number of donor cells from different lymphoid organs was identified by flow cytometry. The resident ratio of CMFDA-positive cells/CMTMR-positive cells in each organ was calculated. Results were expressed as the homing index (resident ratio/input ratio).

**Flow chamber assay.** Flow chamber assays were performed as described[10,11,14,52,53]. Biotinylated 2-GSP-6 (500 ng ml$^{-1}$) or 6-sulfo-sLe$^x$ (1 mg per ml, provided by Dr Nicolai Bovin, Shemyakin-Ovchinnikov Institute of Bioorganic Chemistry) was captured by streptavidin (Pierce) pre-adsorbed to a demarcated region in a 35-mm polystyrene dish. Mouse P-selectin-IgM or E-selectin-IgM was captured by using goat antibody against the Fc portion of human IgM (10 μg ml$^{-1}$) pre-adsorbed on the dish. In some experiments, 20 μg ml$^{-1}$ mouse

ICAM-1-Fc chimera and 2 μg ml$^{-1}$ mouse CXCL1 were also adsorbed. After blocking with 1% HSA, mouse BM or PB leukocytes ($10^6$ per ml in HBSS containing 1.26 mM Ca$^{2+}$, 0.81 mM Mg$^{2+}$ and 0.5% HSA) were perfused over the coated surfaces. In some experiments, 20 μg per ml anti-P-selectin mAb, anti-E-selectin mAb, anti-ICAM-1 mAb, anti-β2 integrin mAb or isotype control mAb were added to the cell suspension. After 5 min, microscopic images of cells under flow were recorded and analysed off-line using a digital analysis system (NIS-Elements, Nikon). Velocities of rolling cells were measured over a 5-s interval.

**Leukocyte aggregation assays.** For static assays, BM cells were flushed with cold HBSS$^-$/EDTA/HSA. After lysis of erythrocytes, cells were resuspended in HBSS with Ca$^{2+}$ and Mg$^{2+}$ (HBSS$^+$) at $4 \times 10^6$ per ml, immediately transferred to a 35-mm dish pre-coated with 1% HSA, and viewed through a microscope with a monitor connected to a CCD digital camera (Dxm1200; Nikon). In other experiments, cells were resuspended in HBSS$^-$/EDTA/HSA or with HBSS$^+$ containing 20 μg ml$^{-1}$ of Mel-14, GAME-46, 4RA10 or isotype control mAb.

For flow assays, isolated BM cells in HBSS$^-$ containing 0.5% HSA were labelled with fluorescent dye PKH67 (green) or PKH26 (red) (Sigma-Aldrich)[53]. Leukocytes labelled each colour ($2 \times 10^6$ per ml) were simultaneously perfused over a dish coated with 1% HSA at a wall shear stress of 1 dyn cm$^{-2}$. Immediately before perfusion, CaCl$_2$ and MgCl$_2$ were added to the suspensions to achieve final concentrations of 1 mM each. In some experiments, 20 μg per ml Mel-14 was added to the suspensions. Cell aggregates in free flow were visualized with a microscope connected to a high-speed camera that captured images at 250 frames s$^{-1}$. Samples were collected after exiting the flow chamber, fixed with 1% paraformaldehyde and immediately analysed by flow cytometry without gate selection[14].

**Intravital microscopy.** Intravital video microscopy of mice anaesthetized by intraperitoneal injection of Avertin (#T48402, Sigma-Aldrich) was performed as described[52,53]. In brief, the cremaster muscle was isolated and superfused with thermocontrolled (35 °C) bicarbonate-buffered saline. Microscopy was performed immediately after isolation (surgical trauma) or 3 h after intrascrotal injection of mouse TNF (R&D Systems, 0.5 μg per mouse in 0.3 ml sterile saline). Visualization was performed under a Nikon Eclipse microscope (E600FN) with a × 40 (NA 0.75) water-immersion objective, coupled to a digital camera (ORCA-Flash 2.8, Hamamatsu) connected to a computer with NIS-Elements software (Nikon). The microscope was also equipped with a Nikon mercury lamp and 510–560-nm excitation filter for acquisition of fluorescent images. Centreline blood flow velocity was measured using a dual photodiode and a digital online cross-correlation program (Microvessel Velocity OD-RT, CircuSoft Instrumentation LLC). Mean blood-flow velocities were obtained by multiplying the centreline velocity by an empirical factor of 0.625. Blood samples (50 μl each) were taken from the carotid catheter, and leukocyte concentrations were measured with the Hemavet 950. Microscopic observations were made up to 1 h for trauma and 2 h for TNF after exteriorization of the cremaster. Three to six venules between 20 μm and 50 μm in diameter per mouse were recorded before and/or after each treatment and saved for off-line analysis. Single or sequential injections of blocking anti-mouse L-selectin mAb Mel-14F(ab′)$_2$, anti-mouse P-selectin RB40.34 or anti-mouse E-selectin mAb 9A9 (30 μg each in 100 μl saline) were administered intravenously. New measurements were made at least 5 min after each injection. The number of leukocytes rolling per minute past a line perpendicular to the vessel's longitudinal axis was counted, and the leukocyte rolling flux fraction was calculated as described[48]. Average rolling velocities of ten leukocytes per venule were measured over a 3-s time window. Firmly adherent leukocytes were defined as those that did not move for at least 30 s and were normalized by surface area for comparison between groups. Emigrated cells were counted in regions of interests extending 50 μm to each side of a vessel over a vessel length of 173 μm and are presented as cells per $10^4$ μm$^2$ tissue area.

**Plasma cytokine/chemokine assay.** Blood was taken from the retro-orbital plexus using EDTA as anticoagulant. Plasma levels of 23 cytokines and chemokines were measured using a Bio-Plex Pro mouse cytokine 23-plex assay kit (#M60009RDPD, Bio-Rad Laboratories, Inc.) in a 96-well plate.

**Neutrophil function assays.** Flow cytometric analysis of respiratory burst and phagocytosis in neutrophils was performed as described[54] with some modifications. For respiratory burst induced by PMA, 100 μl of blood was lysed, washed and resuspended in 500 μl HBSS$^-$/EDTA/HSA. After incubation with 2 μM dihydrorhodamine123 (DHR123, Molecular Probes) for 10 min at 37 °C, PMA (0 to 1 μM, Sigma-Aldrich) was added and incubated for 15 min at 37 °C. The reactions were stopped by placing the tubes on ice, and conversion of DHR123 to fluorescent R123 was analysed by flow cytometry. For respiratory burst induced by phagocytosis of live *Escherichia coli* (*E. coli*, ATCC 25922), 100 μl of heparinized whole blood was incubated with 50 μM dihydroethidium (DHE, Molecular Probes) for 10 min at 37 °C. Samples were then placed on ice. *E. coli* grown overnight at 37 °C in LB medium were washed, resuspended in ice-cold HBSS$^+$ and added to each sample to a final concentration of $2 \times 10^8$ colony-forming units per ml. Mixtures were placed on ice for 15 min and then incubated at 37 °C for 10 or

30 min. Following incubation, erythrocytes were lysed, and cells were resuspended in cold HBSS⁻/EDTA/HSA and analysed by flow cytometry.

The phagocytic capacity of neutrophils was determined with Alexa Fluor 488-labelled *E. coli* (*E. coli*-AF488, Invitrogen). Briefly, 100 µl of heparinized whole blood was cooled on ice and mixed with *E. coli*-AF488 (20 µl, $6 \times 10^5$ bacteria). Mixtures were incubated on ice for 15 min and then incubated at 37 °C for indicated periods or left on ice as control. Phagocytosis was stopped by placing the tubes on ice. Anti-Ly6G-PE was added to label the neutrophil population, and samples were incubated on ice for 30 min. Then, 25 µl of 0.4% trypan blue was added to quench the fluorescence of surface-bound bacteria. After lysis of erythrocytes, the remaining cells were resuspended in cold HBSS⁻/EDTA/HSA. The percentage of neutrophils undergoing phagocytosis and the mean fluorescence intensity (MFI) were determined by flow cytometry. The phagocytic capacity was expressed as the phagocytosis index: (MFI × % positive cells) at 37 °C minus (MFI × % positive cells) at 0 °C.

Ten thousand events were collected in a neutrophil gate identified on a forward scatter-side scatter plot and/or by the neutrophil marker Ly6G using a FACSCalibur flow cytometer. The results are reported as the increase in the mean fluorescence (FL1 for DHR123 and AF488; FL2 for DHE) of the gated cells.

Acute *E. coli*-infected peritonitis was used to evaluate the bacterial killing efficiency of neutrophils *in vivo*[55]. Briefly, *E. coli* in LB medium were sub-cultured at 37 °C to logarithmic growth from an overnight culture. Bacteria were washed and resuspended in saline. The concentration was estimated by absorbance at 600 nm using a predetermined calibration curve and confirmed by viable bacterial counts on LB plates. Mice were injected intraperitoneally with $2 \times 10^7$ colony-forming units of *E. coli* in 0.5 ml saline. After 4 h, blood was collected by retro-orbital puncture. Mice were killed, and the peritoneal cavity was lavaged with 10 ml of cold HBSS⁻/EDTA/HSA. Lavage fluid was serially diluted on LB plates, and the surviving bacteria were enumerated after overnight culture. Body temperatures of each mouse were measured with a rectal probe (MicroTherma 2T, Thermoworks) at the time of bacterial injection and just before the killing.

**Bone marrow transplantation.** Recipient mice were lethally irradiated (9.5 Gy) with a Mark I cesium-137 irradiator[56]. Donor bone marrow cells ($2 \times 10^6$ unfractionated nucleated cells in 200 µl saline) from LysM-GFP or N138G/LysM-GFP mice were injected intravenously into recipient WT, FT DKO or N138G/FT DKO mice. After transplantation, mice were maintained on autoclaved water with antibiotics and were fed autoclaved food. Experiments were performed 6–8 weeks after transplantation once haematopoiesis was reconstituted.

**Neutrophil recruitment during sterile inflammation.** Thioglycollate-induced peritonitis was performed as described[53]. Briefly mice were injected intraperitoneally with 1.5 ml of 4% thioglycollate (Becton Dickinson). After 2, 4, 8, 16 or 24 h, mice were killed and the peritoneal cavity was lavaged with 6 ml ice-cold HBSS⁻/EDTA/HSA.

Neutrophil chemotaxis into air pouches was performed as described[57]. Briefly, 3 ml of filtered sterile air was injected subcutaneously into the dorsal nuchal region on day 0 and day 3. On day 6, each air pouch was injected with 1 ml saline with or without 0.5 µg mouse recombinant CXCL1 or CXCL2 (R&D Systems). After 4 h, the mice were killed and the air pouches were lavaged twice with 1 ml ice-cold HBSS⁻/EDTA/HSA.

The total cells lavaged from peritoneal cavities or air pouches were counted with a haemocytometer. The percentage of neutrophils in the lavage was determined by flow cytometry based on Ly6G expression. The total number of neutrophils recovered was calculated.

Croton oil-induced dermatitis was performed as described[52]. Briefly, mice anaesthetized by intraperitoneal injection of Avertin were treated topically with 10 µl of 2% croton oil (TCI America) in 4:1 acetone: olive oil on each side of the right ear and with the same amount of vehicle control to the left ear. Ear thickness was measured using a digital micrometer (Model:227-211, Mitutoyo Corporation) before and at each indicated time point after application. Ears were removed, fixed in 10% formalin, embedded in paraffin, sectioned and stained with haematoxylin and eosin for light microscopy. Emigrated neutrophils were assessed by counting myeloperoxidase-positive cells stained with rabbit polyclonal antibody against human and mouse myeloperoxidase (Thermo Scientific).

**Plasma clotting time.** Plasma clotting time was assayed as described[58]. Briefly, blood was collected by cardiac puncture in 10% acid-citrate-dextrose buffer. Platelet-poor plasma was prepared by sequential centrifugations at 1,500 *g* for 25 min and 15,000 *g* for 2 min. Plasma was re-calcified by adding an equal volume of calcium chloride (25 mM). The time taken to clot was measured using a Stago ST4 coagulometer in the presence or absence of 100 µg per ml rabbit anti-tissue factor IgG (American Diagnostica Inc. #4515).

**Deep vein thrombosis model.** Deep vein thrombosis induced by flow restriction of the IVC was performed as described[34], with minor modifications. During the surgical procedure, mice were anaesthetized by continuous inhalation of 1–2% isoflurane in 60% oxygen using a veterinary vaporizer and placed on a heating pad in a supine position. After laparotomy, intestines were exteriorized and covered

with saline-moistened gauze to prevent drying. The IVC was gently separated from the aorta just below the left renal veins. Permanent ligation of the IVC over a 30-gauge needle (outside diameter of 0.31 mm serving as a spacer) placed on the vessel was achieved with a 7.0 nylon, non-absorbable suture (Braintree Scientific, Inc.). The needle spacer was then removed to prevent complete IVC occlusion. Side branches were not ligated. The peritoneum and skin were then closed. The entire procedure was completed within 10–15 min. Mice were killed after 48 h, and the IVC was excised below the ligation and proximal to the confluence of the common iliac vein. Thrombus formed in the IVC was removed for measurement of weight and length, for histological analysis, and for western blots. Some mice were injected retro-orbitally with 50 µg Mel-14 F(ab′)₂ immediately before and 24 h after surgery. Preliminary experiments demonstrated that this dose was sufficient to block L-selectin-dependent rolling for at least 24 h.

**Immunostaining of tissue sections.** Neutrophil emigration during croton oil-induced skin inflammation was quantified by myeloperoxidase staining. Sections of ear skin (5-µm) were deparaffinized, rehydrated and treated with heat-induced antigen retrieval in unmasking solution (Vector Laboratories), followed by incubation in blocking buffer (Dako) for 1 h at room temperature. Sections were incubated with rabbit anti-myeloperoxidase IgG or control rabbit IgG overnight at 4 °C. Endogenous peroxidase was blocked in 3% $H_2O_2$ in PBS for 10 min before addition of goat horseradish peroxidase-conjugated anti-rabbit IgG for 1 h at room temperature. The signal was developed using a diaminobenzidine substrate kit (BD Pharmingen). Sections were counterstained with haematoxylin. Images were taken at × 63 magnification. Myeloperoxidase-positive cells were counted in ten adjacent fields of each section.

Neutrophils in thrombi were quantified by immunofluorescence staining of frozen sections. Thrombi were isolated and fixed in 4% paraformaldehyde overnight at 4 °C, transferred into 20% sucrose overnight at 4 °C, embedded in Tissue-Tek O.C.T. Compound (Triangle Biomedical Sciences, Inc.), and processed into 8-µm sections. After fixation and permeabilization in acetone at − 20 °C for 2 min, cryosections were rinsed with PBS containing 0.01% saponin, incubated with serum-free protein block (Dako) at room temperature for 60 min, and then incubated with 5 µg per ml anti-Ly6G mAb containing 0.01% saponin overnight at 4 °C. The tissue sections were stained with Alexa Fluor 488-conjugated donkey anti-rat IgG (5 µg per ml) containing 0.01% saponin at room temperature for 1 h. Nuclei in some sections were counterstained with DAPI. After washing, mounting medium was added to the slides. Sections were visualized on a Zeiss Eclipse 80i microscope. Ly6G-positive percentages were quantified with ImageJ (NIH). Briefly, × 10 images of whole thrombi were obtained by stitching partial images together. The stitched images were converted into 8-bit files and binarized. Using the 'analyse particles' function in ImageJ, the total Ly6G-positive area was divided by the total area of the thrombus section to calculate the percentage area occupied by Ly6G-positive cells.

**Western blots.** Thrombi collected 48 h after stenosis of the IVC were minced and lysed in 1% Triton X-100, 125 mM NaCl, 50 mM Tris pH 8.0, 0.1% SDS, 10 mM EDTA and 1:100 dilution of protease inhibitor cocktail (Thermo Scientific). The lysates were vortexed and centrifuged at 10,000 r.p.m. for 30 min at 4 °C. Aliquots of supernatant, each containing 150–200 µg protein, were resolved by SDS–PAGE under reducing conditions, transferred to PVDF membrane and probed with rabbit anti-Ly6G IgG (Biorbyt, #orb13552, 1:1,000 dilution) or rabbit anti-human fibrin IgG (crossreacts with mouse fibrin) (Dako, #A0080, 1:2,000 dilution), followed by goat horseradish peroxidase-conjugated anti-rabbit IgG (Cell Signaling Technology, #70745, 1:2,000 dilution). Bound antigen was quantified by densitometry[53]. The data are presented as the densitometric ratio of Ly6G to the fibrin Aα chain.

**Statistical analysis.** Sample sizes were determined based on the investigators' qualitative assessment of the reproducibility and variability of each experiment. The experiments were not randomized. The investigators were blinded to the group allocations during experiments and data analysis. Differences between groups were analysed using the two-sided Student's *t* test. Thrombus frequencies were analysed using $\chi^2$ tests of contingency tables. Values were considered significant at $P < 0.05$.

**Data availability.** The data that support the findings of this study are available from the corresponding author on request.

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

## Acknowledgements

We thank Cindy Carter for technical assistance, Nicolai Bovin for providing biotinylated 6-sulfo-sLe$^x$, Jonathan Homeister and Nancy Ruddle for providing mice, and Preston Larson for assistance with scanning electron microscopy. This work was supported by National Institutes of Health Grants AI077343, HL034363, HL128390 and HL085607.

## Author contributions

Z.L., T.Y., N.Z., S.R.P., Y.W., L.Y. and P.M.-D. conducted experiments. Z.L, C.Z., L.X. and R.P.M. analysed the data. Z.L. and R.P.M. designed experiments and wrote the paper, with input and final approval from all authors.
