## [Peer Review File · Nature Communications]

Reviewers' comments:

Reviewer #1 (Remarks to the Author):

This is a well done and very interesting manuscript that studied mice expressing a mutant L-selectin (N138G KI) that was predicted to function in vivo in leukocytes like a slip bond rather than a catch bond (typical of WT L-selectin) during adhesive interactions. L-selectin functions in naïve T/B cell homing in lymphoid tissues and in monocyte and neutrophil recruitment via primary and secondary tethers in vivo and in vitro. The key findings were that the mouse neutrophil's were primed as they enter the circulation, exhibited "unstable aggregates" that lead to greater levels of bacteria phagocytosis and greater inflammation with excessive tissue damage. Interestingly, the authors reported no defect in basal lymphocyte homing or neutrophil recruitment to sites of inflammation.

Some clarifications/ are needed. Manuscript would also be significantly strengthened by testing whether N138G KI mice have a normal response to a well characterized antigen (point 2).

1. Page 10/top page 11. I agree with the authors data priming of PB leukocytes was likely due to adhesive interactions with EC expressed L-selectin ligands. The authors document BM leukocytes are primed as detected by assay of surrogate markers of activation. Can the authors address what causes priming of BM neutrophils and other leukocytes in the BM prior to their release into peripheral blood?

2. Is it very surprising that although L-selectin catch bond formation does not occur in mutant lymphocytes, their homing to LN is normal. This is not consistent with the literature, and should be investigated to a deeper level to substantiate mutant lymphocytes can generate an Ag specific response. L-selectin on lymphocytes is crucial for their entry to LN and for proper immune surveillance, and lymphocyte-HEV interactions are presumed to require L-selectin formation of catch bonds with the HEV ligands. The authors' data imply catch bond formation is not necessary during lymphocyte trafficking to LN. The author should demonstrate that mutant T cells can generate an immune response to Ag (Ova immunization and production of Th1/Th2/Treg/Th17 subsets and positive Ag recall response). This would confirm that an Ag response can occur in these animals with mutant L-selectin.

Reviewer #2 (Remarks to the Author):

The authors constructed mice in which L-selectin bonds are not catch bonds (N138G knockin). They conducted elegant experiments in vitro and in vivo with mixed chimeras and multiple selectin ligand manipulations. These mice showed some neutrophil priming (elevated Mac-1, lower L-selectin surface expression) and premature aggregation. The authors report two clinically relevant outcomes: enhanced bacterial phagocytosis and larger deep vein thrombi in the N138G mice. In general, the data is complete and of high quality. The findings are interesting and novel.

1. Figure 1: What were the assumptions used to calculate the tether force? How were 2-GSP-6 and 6-sulfo-sLex anchored? The authors say a low density was used. How low? And what is the evidence that these molecules stay monomeric? On high density glycans, rolling was restricted to neutrophils. This is not easy to understand, since naïve T cells and classical monocytes express L-selectin at levels similar to neutrophils. The authors clearly have intravital microscopy capabilities (figure 2g-j). Why not show altered rolling velocities in vivo (figure 1)?

2. Figure 2: these aggregates should also be reversed by PSGL-1 mAb.

3. Figure 3 a and b: Was the LFA-1 expression in N138G neutrophils normal?

4. Figure 4: the FT DKO mice are an impressive control and confirm the proposed mechanism.

5. The data from the sepsis model (figure 5) clearly show that this is a relevant mechanism.

6. Figure 7: Impressive and conclusive data. The source of TF+ microparticles is probably monocytes, not neutrophils.

7. In the discussion, the authors say "The persistence of priming in neutrophils that express L-selectinN138G but not PSGL-1 excludes a postulated L-selectin-PSGL-1 signaling complex." Ref. 42 only shows that PSGL-1 signaling requires L-selectin, but not the L-selectin signaling requires PSGL-1. In the present study, L-selectin was mutated and altered signaling was found. PSGL-1 signaling was not tested. In fact, based on the negative data in figure 4e, the authors conclude that PSGL-1 signaling was not involved.

Minor

1. Figure 6: panels mentioned out of sequence.

Reviewer #3 (Remarks to the Author):

In the present manuscript, Lui et al. introduce a novel mouse model to elucidate the mechanochemistry of L-selectin and its impact on neutrophil function in vivo. The authors provide a well-controlled and carefully conducted study of highest quality, reporting that the N138G substitution in L-selectin changed the mechanochemistry of L-selectin by extending bond lifetimes and abolished the characteristic biphasic pattern for the transition from catch to slip bonds. As expected, the number of rolling mutant neutrophils was increased and their rolling velocity was decreased at low shear compared to WT cells. Mutant neutrophils formed homotypic L-selectin-mediated aggregates in vitro and in vivo during leukocyte rolling. Interestingly, only circulating neutrophils but not cells from the bone marrow were primed as shown by an increase in ROS production. The priming resulted in enhanced bacterial killing but also in delayed healing after sterile injury and augmented the size of venous thrombi in a vena cava ligation model. Additionally, priming of L-selectin mutant neutrophils was prevented in fucosyltransferase IV and VII double knockout mice but not in PSGL-1- or lymphotoxin- α -deficient mice (showing little or no expression of the selectin ligand PNAd) indicating that the effects did not directly depend on PSGL-1 or PNAD. Experiments using chimeric mice revealed that priming signals were transmitted via L-selectin itself. In summary, the authors provide an elegant study with novel and convincing data on the mechanochemistry of L-selectin and unraveled its functional importance for neutrophil biology in vivo that merit publication in Nature Communications.

Major concerns:

1. The L-selectinN138G bonds showed slip bound behavior at all shear conditions tested. Can higher shear stress (e.g. > 10 dyn/cm²) induce catch bonds?
2. How would the authors explain the difference in human L-selectinN138G that showed catch bonds at the same force range compared to the tested murine L-selectin mutant?
3. Regarding the massive aggregation of N138G neutrophils observed in in vitro experiments: can the authors prevent this by adding red blood cells according to their hypothesis that this inhibits massive aggregation in vivo? Is aggregation in vitro affected in the presence of shear stress?
4. Authors suggested that healing was impaired in N138G mice after sterile ear injury. Is this finding also true for other models of inflammation? Is it possible that the observed effects are due to delayed resolution of inflammation caused by excessive neutrophil activation?
5. Did the authors test for effects on neutrophil post-adhesion events during neutrophil recruitment in vivo in N138G mice?

Minor points:

1. Correct the sentences on page 3, passage 2, line 3 and 4: Selectins initiate... and Rolling

neutrophils integrate...

2. Page 8, description of cremaster muscle preparation: please use the word "trauma" in the text for clarity as indicated in figure 2.
3. Page 18, method section, reagents: please specify which kit was used for preparation of Mel-14 F(ab')₂ fragments.
4. Figure 2, g-h: please present microscopic images of WT mice for comparison.
5. Figure 6, e: please present microscopic images of neutrophil emigration into ear skin.
6. Figure 6, f: please present the vehicle/left ear of N138G mice for comparison.
7. Figure 7, a: the error bars and statistics are missing.
8. Figure 7, e: please increase brightness or contrast of the image. The fluorescence is hard to see.
9. Supplementary Figure 1: introduce (e) into the figure as mentioned in the figure legend

REVIEWERS' COMMENTS:

Reviewer #1 (Remarks to the Author):

The authors have addressed my comments and i have no further concerns or comments. Its a very solid set of data.

Reviewer #2 (Remarks to the Author):

adequately revised

Reviewer #3 (Remarks to the Author):

I agree with the author's revision of the manuscript.

Point-by-point response to reviewers

We thank the reviewers for their positive comments and their constructive suggestions. Below we respond to each issue raised.

Reviewer #1 (Remarks to the Author):

This is a well done and very interesting manuscript that studied mice expressing a mutant L-selectin (N138G KI) that was predicted to function in vivo in leukocytes like a slip bond rather than a catch bond (typical of WT L-selectin) during adhesive interactions. L-selectin functions in naïve T/B cell homing in lymphoid tissues and in monocyte and neutrophil recruitment via primary and secondary tethers in vivo and in vitro. The key findings were that the mouse neutrophil's were primed as they enter the circulation, exhibited "unstable aggregates" that lead to greater levels of bacteria phagocytosis and greater inflammation with excessive tissue damage. Interestingly, the authors reported no defect in basal lymphocyte homing or neutrophil recruitment to sites of inflammation.

Some clarifications/ are needed. Manuscript would also be significantly strengthened by testing whether N138G KI mice have a normal response to a well characterized antigen (point 2).

Please see response to point 2 below.

1. Page 10/top page 11. I agree with the authors data priming of PB leukocytes was likely due to adhesive interactions with EC expressed L-selectin ligands. The authors document BM leukocytes are primed as detected by assay of surrogate markers of activation. Can the authors address what causes priming of BM neutrophils and other leukocytes in the BM prior to their release into peripheral blood?

BM cells from N138G mice exhibited mild priming as assessed by slightly lower surface levels of L-selectin and, in myeloid cells, slightly higher surface levels of Mac-1. The cause for this is unclear. Unlike PB cells, BM cells from N138G mice did not produce more ROS in response to a second agonist (Fig. 3e).

2. It is very surprising that although L-selectin catch bond formation does not occur in mutant lymphocytes, their homing to LN is normal. This is not consistent with the literature, and should be investigated to a deeper level to substantiate mutant lymphocytes can generate an Ag specific response. L-selectin on lymphocytes is crucial for their entry to LN and for proper immune surveillance, and lymphocyte-HEV interactions are presumed to require L-selectin formation of catch bonds with the HEV ligands. The authors' data imply catch bond formation is not necessary during lymphocyte trafficking to LN. The author should demonstrate that mutant T cells can generate an immune response to Ag (Ova immunization and production of Th1/Th2/Treg/Th17 subsets and positive Ag recall response). This would confirm that an Ag response can occur in these animals with mutant L-selectin.

The literature does not address whether L-selectin catch bonds are required for lymphocyte homing. Indeed, no previous study examined the role of selectin catch bonds or more broadly, selectin mechanochemistry, *in vivo*. Our data establish that altering L-selectin catch bonds by prolonging bond lifetimes at low forces does not impair lymphocyte homing in N138G mice. This is not surprising: *in vitro*, N138G leukocytes rolled normally on L-selectin ligands at physiological shear stresses.

We demonstrate that altered L-selectin mechanochemistry in N138G mice induces priming of leukocytes after they enter the circulation. Our manuscript focuses on how priming of neutrophils affects innate immunity; this required extensive experimentation both *in vitro* and *in vivo*. How priming of naïve and memory/effector lymphocytes affects adaptive immunity is an intriguing topic for future study. However, this will also require extensive experimentation, not merely the proposed Ova immunization protocol. We respectfully submit that this is outside the scope of the current manuscript.

Reviewer #2 (Remarks to the Author):

The authors constructed mice in which L-selectin bonds are not catch bonds (N138G knockin). They conducted elegant experiments in vitro and in vivo with mixed chimeras and multiple selectin ligand manipulations. These mice showed some neutrophil priming (elevated Mac-1, lower L-selectin surface expression) and premature aggregation. The authors report two clinically relevant outcomes: enhanced bacterial phagocytosis and larger deep vein thrombi in the N138G mice. In general, the data is complete and of high quality. The findings are interesting and novel.

1. Figure 1: What were the assumptions used to calculate the tether force? How were 2-GSP-6 and 6-sulfo-sLex anchored? The authors say a low density was used. How low? And what is the evidence that these molecules stay monomeric? On high density glycans, rolling was restricted to neutrophils. This is not easy to understand, since naïve T cells and classical monocytes express L-selectin at levels similar to neutrophils. The authors clearly have intravital microscopy capabilities (figure 2g-j). Why not show altered rolling velocities in vivo (figure 1)?

The Results and Methods now cite our original papers demonstrating how tether forces are calculated. The papers also describe how dimeric and monomeric bonds are distinguished. Tether lifetimes were performed on low ligand densities that permitted transient tethers but not rolling. Bond lifetimes measured as transient tethers were validated by independent measurements using atomic force microscopy. Bone marrow leukocytes were used to measure tether lifetimes and rolling velocities. The great majority of these cells are neutrophils and other myeloid cells. Bone marrow lymphocytes, mostly B cells, express low levels of L-selectin. Thus, few of them tethered to or rolled on the L-selectin ligands (Fig. 1). Naïve T cells and classical monocytes in *peripheral blood* do express higher L-selectin levels. However, it is difficult to measure rolling velocities of these cells on L-selectin ligands *in vivo*. This requires intravital microscopy of intravenously injected, fluorescent cells in high endothelial venules of lymph nodes. However, we did measure rolling of splenocytes and peripheral blood

neutrophils on L-selectin ligands in vitro (Supplementary Fig. 4).

2. *Figure 2: these aggregates should also be reversed by PSGL-1 mAb.*

Correct: anti-PSGL-1 mAb does prevent aggregation. These data have been added to Fig. 2a.

3. *Figure 3 a and b: Was the LFA-1 expression in N138G neutrophils normal?*

Yes, please see Supplementary Fig. 5a. This has been clarified in the text.

4. *Figure 4: the FT DKO mice are an impressive control and confirm the proposed mechanism.*

5. *The data from the sepsis model (figure 5) clearly show that this is a relevant mechanism.*

6. *Figure 7: Impressive and conclusive data. The source of TF+ microparticles is probably monocytes, not neutrophils.*

We appreciate the endorsement of these data.

7. *In the discussion, the authors say “The persistence of priming in neutrophils that express L-selectinN138G but not PSGL-1 excludes a postulated L-selectin-PSGL-1 signaling complex.” Ref. 42 only shows that PSGL-1 signaling requires L-selectin, but not the L-selectin signaling requires PSGL-1. In the present study, L-selectin was mutated and altered signaling was found. PSGL-1 signaling was not tested. In fact, based on the negative data in figure 4e, the authors conclude that PSGL-1 signaling was not involved.*

Actually, Ref. 42 does claim that an L-selectin-PSGL-1 complex is required for L-selectin signaling (see Fig. 3E in Ref. 42). We have revised our sentence to clarify that we find no requirement for PSGL-1 in L-selectin signaling. The reviewer is correct that we did not address whether L-selectin is required for PSGL-1 signaling.

Minor

1. *Figure 6: panels mentioned out of sequence.*

We have corrected the sequence.

Reviewer #3 (Remarks to the Author):

In the present manuscript, Lui et al. introduce a novel mouse model to elucidate the mechanochemistry of L-selectin and its impact on neutrophil function in vivo. The authors provide a well-controlled and carefully conducted study of highest quality, reporting that the N138G substitution in L-selectin changed the mechanochemistry of L-selectin by extending bond lifetimes and abolished the characteristic biphasic pattern for the transition from catch to slip bonds. As expected, the number of rolling mutant

neutrophils was increased and their rolling velocity was decreased at low shear compared to WT cells. Mutant neutrophils formed homotypic L-selectin-mediated aggregates in vitro and in vivo during leukocyte rolling. Interestingly, only circulating neutrophils but not cells from the bone marrow were primed as shown by an increase in ROS production. The priming resulted in enhanced bacterial killing but also in delayed healing after sterile injury and augmented the size of venous thrombi in a vena cava ligation model. Additionally, priming of L-selectin mutant neutrophils was prevented in fucosyltransferase IV and VII double knockout mice but not in PSGL-1- or lymphotoxin-?-deficient mice (showing little or no expression of the selectin ligand PNAd) indicating that the effects did not directly depend on PSGL-1 or PNAD. Experiments using chimeric mice revealed that priming signals were transmitted via L-selectin itself. In summary, the authors provide an elegant study with novel and convincing data on the mechanochemistry of L-selectin and unraveled its functional importance for neutrophil biology in vivo that merit publication in Nature Communications.

Major concerns:

1. The L-selectinN138G bonds showed slip bound behavior at all shear conditions tested. Can higher shear stress (e.g. > 10 dyn/cm²) induce catch bonds?

Higher shear stresses progressively shorten the lifetimes of slip bonds. We do not observe transient tethers at shear stresses greater than 3-4 dyn/cm². This is due to both reduced tethering rates and very short tether lifetimes. At ligand densities that support rolling, cells roll progressively faster at higher shear stresses until they detach. Independent measurements with atomic force microscopy or biomembrane force probes detect only slip bonds at higher applied forces.

2. How would the authors explain the difference in human L-selectinN138G that showed catch bonds at the same force range compared to the tested murine L-selectin mutant?

We cannot readily explain the difference. The methodologies used for studies of human and murine L-selectinN138G are similar. With human L-selectinN138G, we see a leftward shift of the biphasic lifetime curve. That is, we still see transitions from catch to slip bonds, but lower forces elicit catch bonds and the lifetimes are longer before they transition to slip bonds. With murine L-selectinN138G, the lifetimes are longer at the lowest forces that we can study and thus only slip bonds are observed. If we could reliably measure lifetimes at even lower forces, we would likely detect transitions from catch to slip bonds for murine L-selectinN138G. Nevertheless, the altered mechanochemistry for human and murine L-selectinN138G is similar: bond lifetimes at low forces are prolonged.

3. Regarding the massive aggregation of N138G neutrophils observed in in vitro experiments: can the authors prevent this by adding red blood cells according to their hypothesis that this inhibits massive aggregation in vivo? Is aggregation in vitro affected in the presence of shear stress?

We found that RBCs to leukocytes at a ratio of 100:1 did not prevent in vitro aggregation. The density of RBCs impaired resolution with standard microscopy, and thus we have not included the data in the revised manuscript. Adding RBCs at a 1000:1 ratio precluded microscopic visualization of leukocytes. Regardless, failure of RBCs to prevent aggregation in vitro would not preclude a role for RBCs in diminishing leukocyte aggregation in circulating blood.

We still see aggregation of neutrophils (suspended in buffer) in vitro in the presence of shear stress (Fig. 2, b and c).

4. Authors suggested that healing was impaired in N138G mice after sterile ear injury. Is this finding also true for other models of inflammation? Is it possible that the observed effects are due to delayed resolution of inflammation caused by excessive neutrophil activation?

We agree that delayed resolution of inflammation is likely to be due to excessive neutrophil activation, since neutrophils entering inflamed sites are already primed (see Fig. 6, c and d). Future studies will address this topic in other models of inflammation.

5. Did the authors test for effects on neutrophil post-adhesion events during neutrophil recruitment in vivo in N138G mice?

We have not examined this topic in detail. Importantly, we have shown that the number of neutrophils entering an inflamed site in several models does not differ in WT and N138G mice. Whether neutrophils respond differently to cues in the extravascular environment requires further study.

Minor points:

1. Correct the sentences on page 3, passage 2, line 3 and 4: Selectins initiate... and Rolling neutrophils integrate...

We are not sure what the reviewer is suggesting. Combining the two sentences would reduce clarity.

2. Page 8, description of cremaster muscle preparation: please use the word "trauma" in the text for clarity as indicated in figure 2.

Done.

3. Page 18, method section, reagents: please specify which kit was used for preparation of Mel-14 F(ab')₂ fragments.

This is now specified in the Methods.

4. Figure 2, g-h: please present microscopic images of WT mice for comparison.

Done.

5. *Figure 6, e: please present microscopic images of neutrophil emigration into ear skin.*

Neutrophil emigration is visible in the sections of croton oil-challenged ears in new Fig. 6e.

6. *Figure 6, f: please present the vehicle/left ear of N138G mice for comparison.*

Presented in new Fig. 6e.

7. *Figure 7, a: the error bars and statistics are missing.*

Error bars are not possible since the frequency data are derived from an exact number of mice, which is now specified in the figure. Thrombus frequencies were analyzed using χ^2 tests of contingency tables, which is now specified in the Methods. Thrombus frequencies between experimental groups were not statistically different.

8. *Figure 7, e: please increase brightness or contrast of the image. The fluorescence is hard to see.*

Done.

9. *Supplementary Figure 1: introduce (e) into the figure as mentioned in the figure legend*

Done.